# ON THE COST-EFFECTIVENESS OF PARTIALLY-ANNOTATING METHODS FOR MULTI-LABEL LEARNING

## ABSTRACT

Precisely annotating instances with multiple labels is costly and has emerged as a significant bottleneck in the real-world multi-label learning tasks. To deal with this problem, the most straightforward strategy is partially-annotating, which aims to reduce the cost by annotating only a subset of labels. Existing works mainly include label-level partially-annotating (LPA), where each instance is assigned a subset of positive labels, and instance-level partially-annotating (IPA), where all positive labels are assigned to an instance, but only a subset of instances are annotated. However, these methods tend to focus on improving model performance under each type of partial annotation, often neglecting a fundamental question: *which method is more cost-effective?* In this paper, we empirically evaluate which partially-annotating method achieves better model performance at the same annotation cost. To make a fair comparison, we manually annotated images in the MS-COCO dataset using two partially-annotating methods and recorded their average annotation time per image. This allows us to train models on two types of partial annotations with the same annotation cost and to compare their performance. Empirical results show that even when the number of examples annotated with IPA is only one-fifth of that of LPA, models trained on IPA annotations significantly outperform those trained on LPA annotations, yielding that IPA is considerably more cost-effective than LPA. To explain its superiority, our causal reasoning framework shows that compared to LPA, IPA preserves complete co-occurrence relationships, enabling the model to capture correlative patterns, which is useful for improving model performance.

## 1 INTRODUCTION

In *single-label* supervised learning, each instance is assigned with only one class label, whereas the real world may be *multi-labeled*, where multiple objects are usually present in a realistic scenario. For example, an image of a street scene often contains objects such as *pedestrians*, *cars*, and *billboards*. Multi-label learning provides a fundamental framework for handling data with multiple semantics by training a MLL model (Liu et al., 2021b). It has been successfully applied to numerous real-world applications such as image annotation (Kuznetsova et al., 2020), medical image analysis (Ihler et al., 2024) and protein subcellular localization (Liu et al., 2022).

Conventional MLL typically assumes that each training instance has been precisely annotated with all of its relevant labels. Unfortunately, it is difficult and costly to collect a large number of precise annotations. To save labeling expense, a feasible solution is to adopt the partially-annotating strategy, *i.e.*, annotating a subset of labels. Specifically, there are two main types of partially-annotating methods, label-level partially-annotating (LPA) and instance-level partially-annotating (IPA). The former annotates all training instances but only a subset of the labels for each instance; while the latter annotates all labels for an instance but only a subset of the instances. Many recent studies have developed specialized learning algorithms for these two types of partial annotations, known as multi-label learning with partial labels (MLLPL) (Durand et al., 2019; Cole et al., 2021) and semi-supervised multi-label learning (SSMLL) (Wang et al., 2020; Xie et al., 2024).

Although these methods have shown improvements in practical performance for their respective learning tasks, a fundamental question remains: *which partially-annotating method is more cost-effective, that is, which achieves more model performance under the same annotation cost?* The answer to this question is not straightforward, as each method has its own advantages. LPA annotates only a subset of labels for each instance, which often results in more examples being annotated. This allows the learner to gain supervised information from a larger number of examples, even if that information is incomplete. In contrast, IPA annotates all positive labels for a subset of examples, enabling it to leverage valuable prior information, such as the class distribution and label correlations, for model training. Therefore, it is crucial to investigate which partially-annotating method offers better cost-effectiveness.

In this paper, we conduct a systematic analysis to study which partially-annotating method is better for annotating a dataset. We consider an extreme LPA method, where each instance is annotated with only one label, allowing us to annotate all instances at the lowest cost. In the practical annotation process, we group all classes into several super-classes and then perform hierarchical annotation, *i.e.*, annotating the super-classes first and then the individual classes, to avoid checking every class. We show that the annotation cost and the value of the annotated labels for the LPA method depend on the randomness in the annotation process. Generally, greater randomness leads to higher annotation costs and increased label values, and vice versa. According to the different levels of randomness in the annotation process, we develop three variants of the LPA method. To make a fair comparison between the LPA and IPA methods, we develop a unified learning framework to train models by incorporating the pseudo-labeling technique, with the goal of leveraging the information of unannotated data. Extensive experimental results verify that models trained on IPA data significantly outperform those trained on LPA data, even when the number of examples annotated with IPA is only one-fifth of that of LPA. To explore the underlying mechanisms, we construct a causal reasoning model that discloses how, compared to LPA, IPA effectively captures co-occurrence relationships during the annotation process, playing a crucial role in enhancing model performance.

## 2 RELATED WORKS

With the development of deep learning, deep neural networks (DNNs) have become the popular learning models for tackling the MLL problems. Existing deep MLL methods can be roughly divided into three groups. In traditional MLL studies, label correlations have been treated as fundamental information for solving the MLL tasks. Building on these findings, the first group of methods focuses on leveraging label correlations through the development of specialized architectures or training strategies. For example, Chen et al. (2019) used a graph convolutional network (GCN) to model the correlation among different labels. The second group aims to improve the commonly used binary cross entropy (BCE) loss function. For example, the ASL loss (Ridnik et al., 2021) incorporates dynamic down-weighting and hard-thresholding techniques to address the issue of positive-negative imbalance in MLL. The last kind of methods aim to improve classification performance by identifying objects of interest. For example, the asymmetric loss (ASL) incorporates dynamic down-weighting and hard-thresholding techniques to address the issue of positive-negative imbalance in MLL. The final group seeks to improve classification performance by identifying objects of interest. A notable example is Query2Label (Liu et al., 2021a), which employs the attention technique to achieve this objective.

Although these methods have demonstrated improvements in the practical performance of MLL, they require that every instance has been annotated with its all relevant labels. To handle this problem, a feasible solution is to adopt cost-effective annotation methods to reduce the annotation cost. Among them, partially-annotating reduces overall costs by annotating only a subset of labels. This approach can be categorized into label-level partial annotation and instance-level partial annotation. Existing works focus on improving model performance under these two types of partial annotation, known as multi-label learning with partial labels (MLLPL) and semi-supervised multi-label learning (SSMLL).

To solve the MLLPL problems, a pioneering method was to design a partial BCE loss (Durand et al., 2019), which down-weights the loss of annotated labels. The subsequent works designed specialized modules to leverage the information from unannotated labels. Pu et al. (2022) developed a unified semantic-aware representation blending framework to recover unknown labels by exploit-

ing instance-level and prototype-level feature representation simultaneously. Chen et al. (2024) proposed to produce pseudo-labels for unknown labels by exploring within-image and cross-image semantic correlations. Sun et al. (2022); Hu et al. (2023) designed positive and negative prompts to adapt the knowledge embedded in the large pretraining model CLIP to MLLPL. Additionally, Shen et al. (2024) proposed an annotation strategy to allow different annotators to focus on different subset of classes, aiming to reduce the annotation workload of each annotator.

An extreme approach of label-level partially annotating is to assign only one label to each instance, and the corresponding problem is referred to as single positive multi-label learning (SPML) (Cole et al., 2021). An intuitive method was to treat unannotated labels as negative and to train a model with the traditional BCE loss. Unfortunately, such a method would introduce a large number of false negative labels, resulting in unfavorable model performance. To solve this problem, several attempts have been made to recovering potential positive labels by either utilizing the regularization (Zhou et al., 2022) or exploiting manifold structure information (Xie et al., 2022).

Compared to MLLPL, there has been relatively limited research aimed at enhancing the performance of SSMLL. Early work (Wang et al., 2020) utilized a deep neural network (DNN) to extract features, followed by a linear model for classification. While this two-stage approach significantly improved SSMLL performance over traditional methods, its lack of end-to-end training constrained further enhancements. To address this limitation, Xie et al. (2024) introduced an end-to-end DNN approach specifically designed for the SSMLL problems. Their method developed a class-distribution-aware thresholding strategy to perform pseudo-labeling in a class-wise manner, thereby eliminating the need to estimate thresholds for each instance.

## 3 DATA ANNOTATION AND MODEL TRAINING

In this section, we begin by introducing some notations, followed by a discussion on how to annotate the data and train the models, ensuring a fair comparison between the two partially-annotating methods.

Let $\boldsymbol{x} \in \mathcal{X}$ be a feature vector and $\boldsymbol{y} \in \mathcal{Y}$ be its corresponding label vector, where $\mathcal{X} = \mathbb{R}^d$ is the feature space and $\mathcal{Y} = \{0, 1\}^q$ is the label space with $q$ possible classes. In the traditional MLL settings, $y_k = 1$ indicates that the $k$-th label is relevant to the instance; while $y_k = 0$, otherwise. In our setting, we use two methods to partially-annotate training data. By using the LPA method, we obtain a partially-labeled dataset $\mathcal{D} = \{(\boldsymbol{x}_i, \hat{\boldsymbol{y}}_i)\}_{i=1}^n$, where $\hat{\boldsymbol{y}}$ is a partial label vector. Here, $\hat{y}_k = 1$ indicates that the $k$-th label is an annotated positive label for instance $\boldsymbol{x}$; while $\hat{y}_k = 0$ indicates the $k$-th label is unannotated. By using the IPA method, we obtain a labeled dataset $\mathcal{D}_l = \{(\boldsymbol{x}_i, \boldsymbol{y}_i)\}_{i=1}^{n_l}$ and an unlabeled dataset $\mathcal{D}_u = \{\boldsymbol{x}_i\}_{j=1}^{n_u}$, where $n_l$ and $n_u$ are the numbers of labeled and unlabeled examples that satisfy $n = n_l + n_u$. Below, we will introduce an annotation engine, which provides an effective tool for annotating data, and an learning engine, which allows to train models based on different annotation methods in a fair manner.

### 3.1 ANNOTATION ENGINE

Given that our empirical studies are mainly based on multi-label benchmark dataset MS-COCO (Lin et al., 2014), which is the most commonly used dataset in MLL literature, we adopt the annotation method that has been applied to annotate the MS-COCO dataset. Following their approach, we first group the original 80 classes into 12 super-classes (see details in Appendix). For a given instance, an annotator is asked to determine whether the instance contains any semantic super-class. This step can significantly reduce the time required to check all the 80 classes. For example, an annotator can easily determine that no *vehicle* is presented in the image of an indoor scene, so the annotator can quickly skip the super-class *vehicle*. When an annotator determines the super-class that an image may belong to, the annotator must check the classes within the super-class one by one and mark the semantic class that is present in the image. Below, we will discuss how to perform the IPA and LPA methods based on the annotation engine.

**Label-Level Partially-Annotating** For the LPA method, to minimize the annotation cost while annotating as many instances as possible, we annotated all instances but provided only one positive label for each instance. Such a problem has been formalized as a learning framework, called single

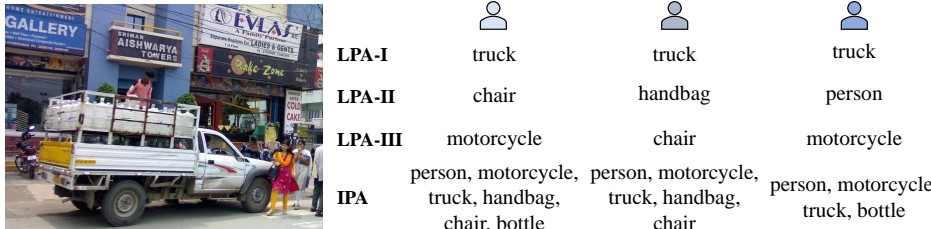

| | ⛑ | ⛑ | ⛑ |
|---|---|---|---|
| **LPA-I** | truck | truck | truck |
| **LPA-II** | chair | handbag | person |
| **LPA-III** | motorcycle | chair | motorcycle |
| **IPA** | person, motorcycle, truck, handbag, chair, bottle | person, motorcycle, truck, handbag, chair | person, motorcycle, truck, bottle |

Figure 1: An example of annotation results using different partially-annotation methods by three annotators. From LPA-I to LPA-III, by increasing the randomness in the LPA method, we can see that the labels provided by the annotators range from the most conspicuous object, like *truck*, to difficult-to-identify objects, like *motorcycle* and *chair*.

positive multi-label learning (SPML) (Cole et al., 2021). Existing works often assume that the single positive label for each instance is annotated randomly. In fact, how positive labels are annotated is an important problem since different annotation methods often result in varying annotation costs and label value. For example, if we annotate only the most prominent object class for each image, *i.e.*, the object that is easiest to annotate, the total annotation cost will be minimized. However, the label value is also minimal, making it difficult for the model to achieve favorable performance. To achieve a balance between annotation costs and label values, we propose the following three LPA methods, which progressively increase the randomness of the obtained labels. These three LPA methods are listed as follows.

- **LPA-I** This method directly annotates the most prominent object class in an image. Although this method can minimize annotation costs, it typically results in less informative labels. As shown in Figure 1, all three annotators provide the label *truck* using the LPA-I method. This object can be easily recognized from the image, resulting in a relatively minor impact on improving model performance.

- **LPA-II** For each image, the annotator is provided with super-classes in a random order by the annotation engine. The task of annotators is to identify the first super-class that the image contains and to annotate the most prominent object class within that super-class. As shown in Figure 1, the randomness of the super-classes may lead annotators to focus on some difficult-to-recognize objects, such as *chair* and *handbag*. These labels are clearly more valuable than *truck* and have a significant impact on improving model performance.

- **LPA-III** For each image, the annotator is provided with super-classes in a random order. The annotator is asked to identify the first super-class contained in the image. Then, the annotator is provided with the classes within the super-class in a random order. The task is to check classes within the super-class in a random order and identify the first object class that is present in the image. By further introducing randomness into the classes, the range of annotations provided by the annotators will continue to expand, increasing label diversity and enhancing model performance. For example, in Figure 1, using the LPA-II method, annotators are unlikely to annotate *motorcycle* because trucks are clearly the easiest objects to annotate within the super-class *vehicle*. In contrast, using the LPA-III method, annotators are likely to annotate *motorcycle* due to the randomness of classes.

Generally, from the perspective of annotation costs, LPA-I has the lowest cost, as it annotates the easiest class each time, while LPA-III has the highest annotation cost because it often requires checking multiple super-classes/classes to obtain the final annotation; from the perspective of the label value, LPA-I provides the least valuable labels because it consistently annotates the most prominent objects, leading to significant information redundancy. In contrast, LPA-III offers the most valuable labels by introducing randomness at both the super-class and class levels, encouraging annotators to focus on difficult-to-recognize objects, which is useful for enhancing model performance.

**Instance-Level Partially-Annotating**    For the IPA method, we progressively annotate each unlabeled with all positive labels based on our annotation engine. Specifically, the annotators must check the super-classes one by one. When they encounter a potential positive super-class, they must check

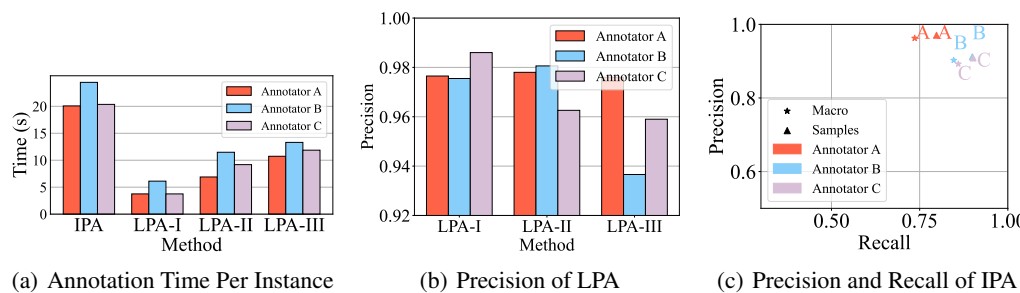

(a) Annotation Time Per Instance  (b) Precision of LPA  (c) Precision and Recall of IPA

Figure 2: Annotation costs and label quality of different partially-annotating methods.

each class within that super-class and annotate the class objects present in the image. The number of annotated examples depends on the total annotation budget, which will be discussed later.

To compare the annotation costs of different partially-annotating methods, three annotators are asked to manually annotate the MS-COCO dataset based on our annotation engine. Considering that the cost of annotating the entire dataset is prohibitively high (requiring a total of four rounds of annotation) and our main goal is to compare the annotation costs, it is not necessary to annotate the entire dataset. For each of the three LPA methods, each annotator is asked to annotate 2,000 images, with each image being annotated a single positive label; for the IPA method, each annotator is asked to annotate 500 images, with each image being annotated with all positive labels. Figure 2(a) shows the annotation costs of the different methods in terms of annotation time per image. From the figure, it can be observed that: i) The annotation cost per image of the IPA method is significantly higher than that of any LPA method, because IPA requires annotating all positive labels for each image, while LPA only requires only a single positive label per image. ii) With the increase of randomness in the annotation process, the annotation cost of the LPA methods will increases accordingly. iii) With the total annotation cost kept constant, the number of annotated examples for the IPA method constitutes 20%, 42%, and 55% of the number of examples annotated by the three LPA methods, respectively.

Furthermore, we analyze the label quality associated with different partially-annotating methods. Figure 2(b) shows the precision of the annotations made by three annotators using the three LPA methods. It is noteworthy that since the LPA method assigns only one label to each instance, the recall metric is not meaningful in this context. From the figure, it can be observed that: i) different LPA methods can achieve high annotation precision. ii) LPA-I and LPA-II achieve comparable precision, while LPA-III exhibits relatively lower precision. One possible reason is that in the LPA-III method, we introduced certain constraints that prevent annotators from always annotating the prominent class object, which increases the difficulty of annotation and thus reduces annotation precision. Figure 2(c) shows the macro/sample precision and macro/sample recall of annotations performed by three annotators using the IPA method. From the figure, we can see that annotating all positive labels for an instance is significantly challenging, and even dedicated annotators can make mistakes, including missed annotations and incorrect annotations.

## 3.2 Learning Engine

To fairly compare two partially-annotating methods, we use a unified learning framework for model training. It can be summarized as two procedures: i) warmup the model on the annotated labels. ii) use the model to generate pseudo-labels for the unannotated parts and retrain the model on all labels. Note that we repeat the second procedure until the model converges.

**Annotated Labels Utilization** For the LPA method, during the warmup stage, it is infeasible to train the model only on the annotated labels, since negative labels are unavailable. To solve this problem, inspired by the previous work (Cole et al., 2021), we adopt the "assume negative" (AN) strategy, which treats all unknown labels as negative. For the IPA method, we train the model directly on the labeled dataset. During the warmup stage, we utilize the commonly used ASL loss, which is an

improved version of the BCE loss, to train the models. Formally, the ASL loss can be defined as

$$\mathcal{L}(f(\boldsymbol{x}), \boldsymbol{y}) = \sum_{i=1}^{q} y_k \ell_1(f_k(\boldsymbol{x})) + (1 - y_k)\ell_0(f_k(\boldsymbol{x})). \tag{1}$$

Here, $\ell_1(f_k) = -(1-f_k)^{\lambda_1} \log(f_k)$ and $\ell_0(f_k) = -(f_k)^{\lambda_1} \log(1-f_k)$ represent losses calculated on positive and negative labels, where $\lambda_1$ and $\lambda_2$ are positive and negative focusing parameters.

**Unannotated Labels Exploration** To leverage the information of unannotated data, we employ the widely used method of pseudo-labeling (PL), which has been proven effective in semi-supervised learning (Berthelot et al., 2019; Sohn et al., 2020; Li et al.). Specifically, after the warmup stage, at each training epoch, we use a teacher model (usually an exponential moving average (EMA) (Tarvainen & Valpola, 2017) of the original model) to produce the predicted probabilities $\boldsymbol{q}$ for all unannotated labels. Then, we use a threshold function $\tau$ to separate these labels into positive and negative ones. Finally, we retrain the model based on pseudo-labels and model predictions $\boldsymbol{p}$. The key question is how to define the thresholding function. Regarding the IPA data, inspired by the recent work (Xie et al., 2024), we use a class-distribution-aware thresholding method to separate the positive and negative labels of unlabeled instances according to the class proportions of the labeled data. Regarding the LPA data, since we have no prior information, it is difficult to set a specific threshold. Generally, a fixed threshold can be used, where the same threshold is applied to all predictions. Specifically, the PL loss function for unannotated labels (denoted by LPA-PL) can be defined as

$$\mathcal{L}_{\text{LPA-PL}}(\boldsymbol{p}, \boldsymbol{q}) = \sum_{k=1}^{q} \mathbb{I}(q_k \geq \tau_{\text{G}} | \hat{y}_k = 0)\ell_1(p_k) + \mathbb{I}(q_k < \tau_{\text{G}} | \hat{y}_k = 0)\ell_0(p_k). \tag{2}$$

The condition $\hat{y}_k = 0$ is used to ensure the loss is only imposed on the unannotated labels. The PL loss function for unannotated instances (denoted by IPA-PL) can be defined as

$$\mathcal{L}_{\text{IPA-PL}}(\boldsymbol{p}, \boldsymbol{q}) = \sum_{k=1}^{q} \mathbb{I}(q_k \geq \tau_{\text{CAT}}^k)\ell_1(p_k) + \mathbb{I}(q_k < \tau_{\text{CAT}}^k)\ell_0(p_k). \tag{3}$$

## 4 EXPERIMENTS

**Dataset** The experiments are mainly performed on benchmark dataset MS-COCO 2014 (MS-COCO for short) [1] (Lin et al., 2014). It consists of 82,081 training examples and 40,504 validation examples, which are categorized into 80 classes. Considering that all examples in the MS-COCO dataset have true labels, we simulated manual annotation using three different LPA methods (LPA-I, LPA-II and LPA-III) to generate three distinct datasets, each with instances that are each assigned a single positive label. For the LPA-III method, which are not affected by the preference of annotators, we can directly use true labels for annotation; for LPA-I and LPA-II methods, which require selecting the most salient object during annotation and are often influenced by the preference of annotators, we simulate this process by selecting the largest object according to sizes of bounding boxes. To fairly compare the IPA method with the three LPA methods, based on the previous obtained ratio of per-instance annotation costs between the different LPA methods and the IPA method, IPA allows us to obtain three datasets, with the proportions of annotated examples being 20%, 42%, and 55%, respectively. For notational simplicity, we denote these three experimental settings as **Mode-I**, **Mode-II** and **Mode-III**.

**Methods** To compare the cost-effectiveness of the LPA and IPA methods, we train models on the LPA and IPA datasets with equivalent annotation costs, and then compared their performance. Specifically, for LPA, we employ two methods: i) **LPA-AN**: training models on LPA data with AN loss by assuming unannotated labels as negative. ii) **LPA-PL**: training models on LPA data using the PL method. For IPA, we employ two methods: i) **IPA-ASL**: training models only on labeled data with the ASL loss. ii) **IPA-PL**: training models on labeled and unlabeled data using the PL method. By comparing LPA-AN and IPA-ASL, we can evaluate which annotation method is more cost-effective when using only the annotated data. By comparing LPA-PL and IPA-PL, we can evaluate which annotation method is more cost-effective when leveraging the information of unannotated data additionally.

---

[1]https://cocodataset.org

Table 1: Comparison results of different models trained on LPA and IPA data on MS-COCO in terms of mAP (%). The image resolution is set as 224 and 448.

| Method | Mode-I | Mode-II | Mode-III | Random | Mode-I | Mode-II | Mode-III | Random |
|---|---|---|---|---|---|---|---|---|
| | ResNet-50, Resolution=224 | | | | ResNet-50, Resolution=448 | | | |
| LPA-AN | 46.47 | 62.42 | 64.93 | 65.53 | 49.05 | 68.04 | 70.69 | 71.34 |
| IPA-ASL | 67.89 | 71.13 | 72.27 | - | 74.37 | 77.59 | 78.66 | - |
| LPA-PL | 46.47 | 62.96 | 65.57 | 65.53 | 49.09 | 68.41 | 71.29 | 71.87 |
| IPA-PL | 69.55 | 72.64 | 73.25 | - | 74.61 | 78.34 | 78.99 | - |
| Supervised | 74.46 | | | | 80.52 | | | |
| | ML-Decoder, Resolution=224 | | | | ML-Decoder, Resolution=448 | | | |
| LPA-AN | 46.83 | 63.91 | 66.27 | 67.11 | 49.67 | 70.11 | 72.76 | 73.53 |
| IPA-ASL | 70.05 | 73.22 | 74.09 | - | 76.79 | 79.54 | 80.37 | - |
| LPA-PL | 47.24 | 64.75 | 67.16 | 67.97 | 50.34 | 70.55 | 73.46 | 74.17 |
| IPA-PL | 70.94 | 74.14 | 74.53 | - | 77.35 | 80.22 | 80.72 | - |
| Supervised | 75.97 | | | | 82.32 | | | |
| | ViT, Resolution=224 | | | | ViT, Resolution=448 | | | |
| LPA-AN | 47.32 | 62.88 | 65.34 | 66.04 | 50.22 | 70.35 | 73.24 | 73.91 |
| IPA-ASL | 64.89 | 66.31 | 67.06 | - | 74.34 | 76.25 | 76.59 | - |
| LPA-PL | 47.32 | 62.88 | 65.34 | 66.04 | 50.22 | 70.35 | 73.24 | 73.91 |
| IPA-PL | 71.42 | 72.57 | 72.78 | - | 79.69 | 80.71 | 80.82 | - |
| Supervised | 73.94 | | | | 82.58 | | | |

Table 2: Comparison results between IPA-PL and the state-of-the-art SPML methods in terms of mAP (%).

| Method | Mode-I | Mode-II | Mode-III | Random |
|---|---|---|---|---|
| PLC | 52.64 | 71.11 | 73.35 | 74.04 |
| BoostLU+LL-R | 51.36 | 69.76 | 72.68 | 73.05 |
| IPA-PL | 74.61 | 78.34 | 78.99 | - |

**Implementation** We employ ResNet-50 (He et al., 2016), ML-Decoder (Ridnik et al., 2023), Visual Transformer (ViT) (Dosovitskiy et al., 2020) as base models. Among them, ResNet-50 and ML-Decoder are pretrained on ImageNet (Deng et al., 2009), and ViT is pretrained on on LAION-2B image-text pairs using OpenCLIP and is Fine-tuned on ImageNet. The widely used RandAugment (Cubuk et al., 2020) and Cutout (DeVries, 2017) techniques are utilized for data augmentation. We employ the AdamW (Loshchilov, 2017) optimizer and the one-cycle policy scheduler to train the model with maximal learning rate of 0.0001. Based on our preliminary experiments, the numbers of warmup epochs is set as 7 and 12 for LPA-PL and IPA-PL methods. The batch sizes are set as 64 for both methods. Furthermore, we perform EMA for model parameters with a decay of 0.9997. The EMA model serves dual purposes: it acts both as a teacher model to generate pseudo-labels for unlabeled data and as a target model for evaluation on test data. We perform all experiments on GeForce RTX 4090 GPUs.

### 4.1 WHAT IS THE BEST METHOD TO ANNOTATE A DATASET?

In this section, we evaluate which annotation method yields better model performance at the same annotation cost. Table 1 reports comparison results of different methods on MS-COCO in terms of mAP by using different base models and image resolutions. From the table, it can be observed that by only training on annotated data, the performance of IPA-ASL is significantly better than that of LPA-AN in all cases. By leveraging the information of unannotated data using the PL method, IPA-PL still significantly outperform LPA-PL with a large margin. In particular, based on ViT models, LPA-PL does not obtain improvements by introducing the PL strategy. One possible reason is that ViT

Table 3: Comparison results of different models trained on LPA and IPA data on MS-COCO in terms of mAP (%). The image resolution is set as 224 and 448.

| Method | Mode-I-N | Mode-II-N | Mode-III-N | Mode-I-N | Mode-II-N | Mode-III-N |
|--------|----------|-----------|------------|----------|-----------|------------|
| | ResNet-50, Resolution=224 | | | ResNet-50, Resolution=448 | | |
| LPA-AN | 46.05 | 61.72 | 63.77 | 48.99 | 67.71 | 70.14 |
| IPA-ASL | 59.38 | 64.98 | 66.66 | 65.01 | 70.73 | 72.55 |
| LPA-PL | 46.14 | 62.31 | 64.65 | 49.19 | 68.09 | 70.64 |
| IPA-PL | 61.52 | 65.26 | 67.01 | 65.87 | 70.73 | 72.69 |

is overfitting to false negative labels, resulting in low-quality pseudo-labels produced by the model. These results convincingly verify that the IPA method can achieve more cost-effective annotation when compared with the LPA method. As mentioned in Section 3.1, although using LPA-I for annotation minimizes the annotation cost, the value of the labels obtained is also the lowest, making it difficult to achieve favorable model performance (in Mode-I). By increasing the randomness in annotation, LPA-II and LPA-III significantly increase label value, enhancing the model performance (in Mode-II and Mode-III). In the Random setting, a label is chosen completely at random to serve as the single positive label. In particular, the model performance in Mode-III (using LAP-III method for annotation) is very close to that of Random. These results validate that increasing the randomness in the LPA method effectively improves the label value of annotations, even though the annotation cost also increases.

As mentioned in Section 3.1, SPML methods can be applied to deal with the LPA data. To mitigate the risk of performance differences caused by the effectiveness of the learning methods, we conduct a further comparison by adopting two state-of-the-art SPML methods, BoostLU (Kim et al., 2023) and PLC (Xie et al., 2022). It is noteworthy that these are the two highest-performing SPML methods for which we can find code available online. Figure 2 reports comparison results between IPA-PL and the comparing methods. From the table, it can be observed that the combination of using the IPA method for annotation and the simple PL method for model training significantly outperforms the combination of using LPA for annotation and state-of-the-art SPML method for model training. These results further validate that IPA is a more cost-effective annotation method than LPA in multi-label image classification tasks.

## 4.2 STUDY ON ANNOTATION NOISE

As discussed in Scetion 3.1, regardless of whether LPA or IPA is used for annotating data, it cannot be guaranteed that the provided labels are completely accurate. In this section, we study a more practical scenario, where the annotated labels contain noise. Specifically, we introduce random noise into the LPA and IPA annotations to simulate label noise that occurs during the annotation process. For LPA, the average precision of manual annotation across the three modes is 97.93%, 97.37%, and 95.72%, with corresponding noise rates of 2.07%, 2.63% and 4.28%. For IPA, the average precision is 93.11% and the average recall is 86.55%, indicating that the noise rates are respectively 6.89% and 13.45%, respectively. We introduce random noise into the three modes according to the noise rates, resulting in three noisy modes, denoted by Mode-I-N, Mode-II-N, and Mode-III-N. Table 3 reports the comparison results between LPA-PL and IPA-PL in terms of mAP. From the table, it can be observed that IPA-PL significantly outperforms LPA-PL in all cases. These results verify that by considering the manual annotation noise, the IPA method is still more cost-effective than the LPA method.

## 5 DISCUSSION

In this section, we discuss the reasons why models trained on the IPA data achieve better performance than that trained on the LPA data under the same annotation cost. The main difference between IPA and LPA is that IPA annotates all labels during the annotation process, automatically introduces prior information of label co-occurrence, whereas LPA only annotates a single label, missing the co-occurrence information among labels. It is noteworthy that even if we annotate a subset of labels for each instance instead of just one label, it still lead to incomplete co-occurrence rela-

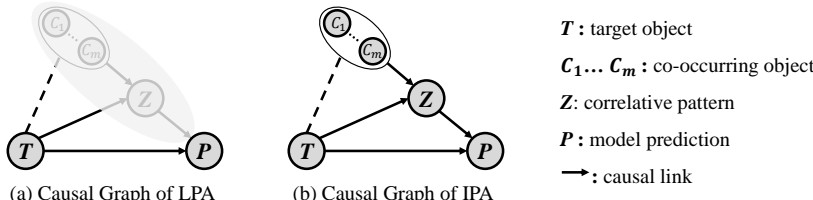

(a) Causal Graph of LPA    (b) Causal Graph of IPA

$T$ : target object

$C_1 ... C_m$ : co-occurring object

$Z$: correlative pattern

$P$ : model prediction

→ : causal link

Figure 3: Causal graphs of LPA and IPA data. For LPA, the prediction is derived based on the path $T \to P$, as the co-occuring objects are uannotated. While the prediction can be dervied on additional path $\{C_1, ..., C_m\} \to Z \to P$.

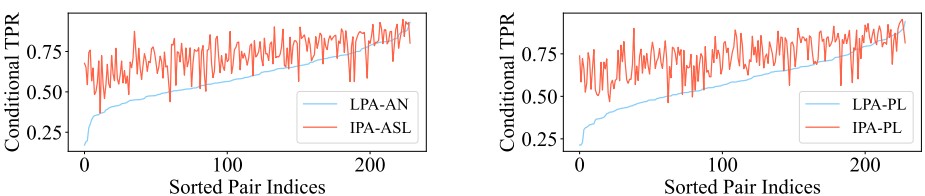

Figure 4: Conditional TPR of the label pairs with co-occurrence probabilities larger than 0.2 on MS-COCO. The pair indices are sorted according to the performance of LPA-AN and LPA-PL, respectively.

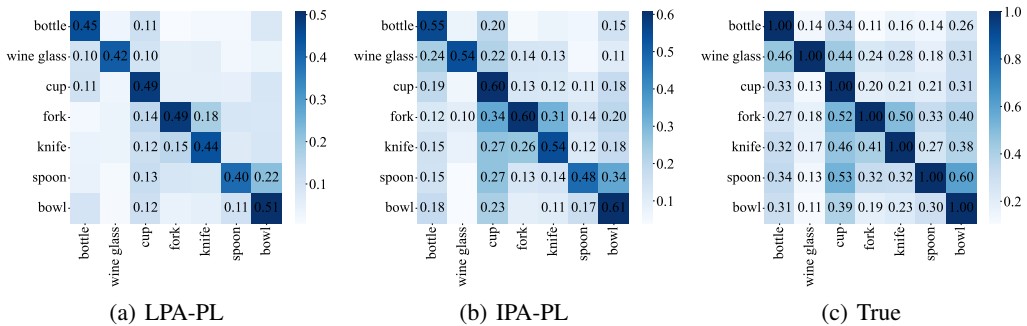

(a) LPA-PL    (b) IPA-PL    (c) True

Figure 5: Co-occurrence matrices of predictions among related classes. Co-occurrence probabilities greater than 0.1 are highlighted along with their values. Compared to LPA-PL, the co-occurrence matrix predicted by IPA-PL is much closer to the true co-occurrence matrix.

tionships. Many studies (Chen et al., 2019; Lanchantin et al., 2021) have shown that co-occurrence relationships are fundamental information for enhancing model performance in MLL. To demonstrate the importance of co-occurrence relationships, we study how co-occurrence affects model predictions from the perspective of causal reasoning. Without loss of generality, we assume that there are $m$ co-occurring objects in an image. Figure 3 illustrate causal graphs involving four variables: the target object $T$, the co-occurring objects $\{C_1, ..., C_m\}$, the correlative pattern $Z$, and the model prediction $P$. Regarding LPA, as shown in Figure 3 (a), the model prediction can only be derived from the path $T \to P$, as the co-occurring objects are not annotated (the masked part). Regarding IPA, different from LPA, the model prediction can be derived from an additional path $\{C_1, ..., C_m\} \to Z \to P$. The additional path significantly enhances the recognition ability on the target object of the model by leveraging the co-occurrence relationships.

To verify the effectiveness of co-occurrence relationships, we report the conditional true positive ratio (TPR) for a specific class given the presence of its co-occurring class. For example, the pair $\{A, B\}$ represents the proportion of images containing both $A$ and $B$, in which the model predicts

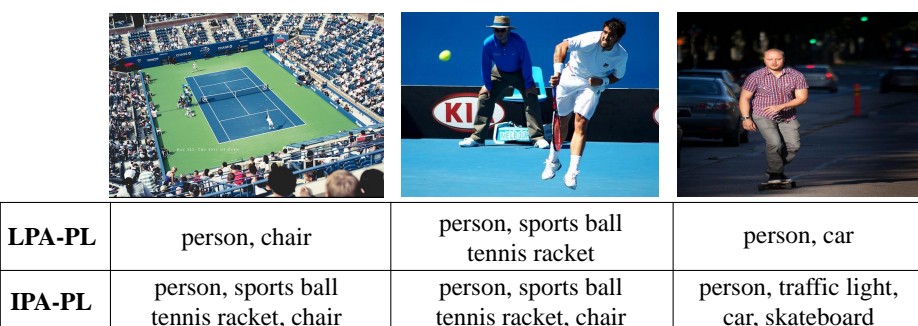

| | | | |
|---|---|---|---|
| **LPA-PL** | person, chair | person, sports ball tennis racket | person, car |
| **IPA-PL** | person, sports ball tennis racket, chair | person, sports ball tennis racket, chair | person, traffic light, car, skateboard |

Figure 6: Visualization of predictions using different methods.

the presence of the object $A$. Figure 4 illustrates the conditional TPR of label pairs with a co-occurrence probability greater than 0.2. For simplicity of presentation, we assign a number to each pair and sort them based on the performance of LPA-AN and LPA-PL. LPA-AN and IPA-ASL are trained with the same configuration (Mode-III, ResNet-50, only annotated data). LPA-PL and IPA-PL are trained on the same configuration (Mode-III, ResNet-50, on both annotated and unannotated data). From the figure, it can be observed that the IPA-ASL and IPA-PL generally outperform LPA-AN and LPA-PL, respectively, yielding that the models are able to leverage the co-occurrence relationships in the IPA annotations for achieving better performance. Figure 5 illustrates the co-occurrence matrices estimated on predictions of different methods among some related classes. The results on some other classes can be found in Appendix. From the figure, we can see that compared to LPA-PL, the co-occurrence matrix obtained from IPA-PL is more consistent with the true co-occurrence matrix. These results disclose that based on IPA annotations, IPA-PL can accurately capture co-occurrence relationships to enhance model performance. To disclose the superiority of IPL-PL on capturing co-occurrence relationships, Figure 6 shows some cases of predictions on MS-COCO. From the figure, it can be observed that by training on the IPA annotations, the model has a greater ability to capture co-occurrence relationships, which in turn enhances its performance in identifying challenging objects, including tiny objects, *e.g.*, *tennis racket, sports ball* in the first image, *traffic light* in the third image, and occluded objects, *e.g.*, *chair* in the second image.

## 6 CONCLUSION

This paper studies an important question: which method is more cost-effective between label-level partially-annotating and instance-level partially-annotating? To answer this question, we conducted a systematic empirical study to compare the annotation cost and label value of these two annotation methods. Regarding the annotation cost, we manually annotate images in MS-COCO using two types of partially-annotating methods and record their averaging annotation time per image. Regarding the label value, we train models on the two types of partially-annotated data under the same annotation budget and compare their performance. Our empirical studies show that IPA achieves better model performance than LPA under the same annotation cost. From the perspective of causal reasoning, we analyze why IPA is a better annotation method: it ensures complete co-occurrence information during the annotation process, which facilitates the learning of co-occurrence patterns, thereby enhancing model performance. In future, we will compare the cost-effectiveness of the two partial-annotating methods on larger-scale datasets.

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

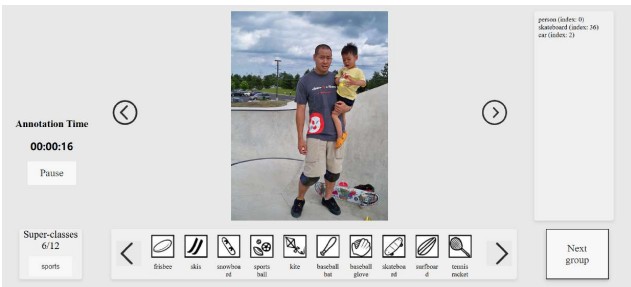

Figure 7: An illustration of the annotation tool used for annotating images in MS-COCO. A total of 80 categories are divided into 12 super-classes. The annotation time is recorded.

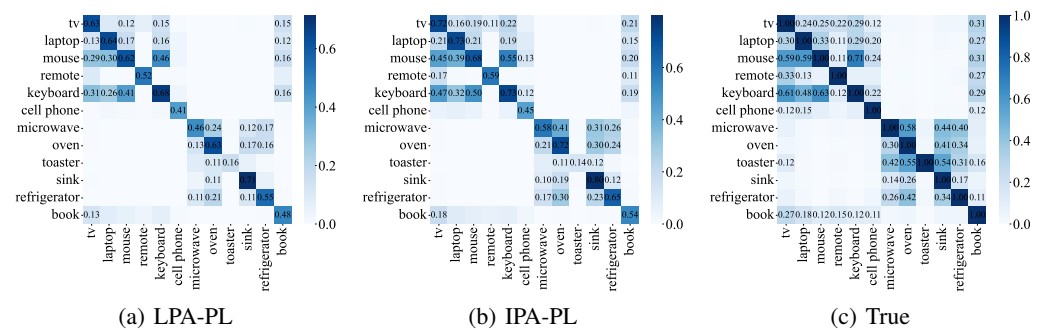

| (a) LPA-PL | (b) IPA-PL | (c) True |
|---|---|---|

Figure 8: Co-occurrence matrices of predictions among some other related classes. Co-occurrence probabilities greater than 0.1 are highlighted along with their values. Compared to LPA-PL, the co-occurrence matrix predicted by IPA-PL is much closer to the true co-occurrence matrix.

Table 4: The division of 12 super-classes

| Super-classes | Classes |
|---|---|
| Person | person |
| Transportation | bicycle, car, motorcycle, airplane, bus, train, truck, boat |
| Outdoor | traffic light, fire hydrant, stop sign, parking meter, bench |
| Animals | bird, cat, dog, horse, sheep, cow, elephant, bear, zebra, giraffe |
| Accessories | backpack, umbrella, handbag, tie, suitcase |
| Sports | frisbee, skis, snowboard, sports ball, kite, baseball bat, baseball glove, skateboard, surfboard, tennis racket |
| Kitchen | bottle, wine glass, cup, fork, knife, spoon, bowl |
| Food | banana, apple, sandwich, orange, broccoli, carrot, hot dog, pizza, donut, cake |
| Furniture | chair, couch, potted plant, bed, dining table, toilet |
| Electronics | tv, laptop, mouse, remote, keyboard, cell phone |
| Appliances | microwave, oven, toaster, sink, refrigerator |
| Indoor | book, clock, vase, scissors, teddy bear, hair dryer, toothbrush |

## A  APPENDIX

Figure 7 provides an illustration of the annotation tool used in our paper. By using this tool, annotators can perform annotation in an hierarchical manner, with the goal of reducing annotation time.

Figure 8 illustrates the co-occurrence matrices of predictions among other related classes. The co-occurrence matrix predicted by IPA-PL is more consistent to the true co-occurrence matrix when compared to LPA-PL.

Table 4 lists the hierarchical relationships between classes and super-classes.

