# OpenReview forum: "On the Cost-Effectiveness of Partially-Annotating Methods for Multi-Label Learning"
_ICLR.cc/2025/Conference — Submitted to ICLR 2025_

### Official Review · Reviewer_HFP9 · 2024-10-16

**Soundness:** 3
**Presentation:** 3
**Contribution:** 3
**Rating:** 6
**Confidence:** 5

**Summary:**

Unlike existing studies, the authors explore an intriguing question in multi-label learning: should we partially annotate at the label level or the instance level? Through extensive experiments and a proposed causal reasoning framework, they demonstrate that instance-level partial annotation (IPA) maintains complete co-occurrence relationships, which proves more beneficial for enhancing multi-label regression (MLR) model performance compared to label-level partial annotation (LPA).

**Strengths:**

1. This work offers a compelling and important motivation, providing essential guidance for future multi-label regression (MLR) research.
2. A comprehensive analysis of extensive experimental results reveals the underlying reasons in detail.

**Weaknesses:**

1. The discussion of MLR-PL is not sufficient, ignoring some recent work (e.g., SARB[1], DualCoOp++[2], HST[3]).
2. Comparison algorithms are somehow outdated.

[1] Semantic-Aware Representation Blending for Multi-Label Image Recognition with Partial Labels, AAAI 2022.
[2] DualCoOp++: Fast and Effective Adaptation to Multi-Label Recognition With Limited Annotations, TPAMI 2023.
[3] Heterogeneous Semantic Transfer for Multi-label Recognition with Partial Labels, IJCV 2024.

**Questions:**

1. Do the paper's conclusions still apply to other contemporary algorithms, e.g., SARB[1], DualCoOp++[2], HST[3]?

---

> ### Author Response · Authors · 2024-11-20
>
> Thanks for your constructive comments. We are glad that you considered our work “offers a compelling and important motivation”. We are glad to solve all your concerns.
>
> **Q1**: The discussion of MLR-PL is not sufficient, ignoring some recent work (e.g., SARB[1], DualCoOp++[2], HST[3]).
>
> **A1**: Thanks for your suggestion. We have included the discussion of these relevant works in the rebuttal version and marked them as blue.
>
> **Q2**: Comparison algorithms are somehow outdated.
>
> **A2**: Our focus is to compare the cost-effectiveness between LPA and IPA, specifically by comparing the performance of models trained on LPA and IPA annotations. To make a fair comparison, we adopted the simple pseudo-labeling method for training on these two annotations. Note that directly comparing the SOTA methods in their respective fields (SPML methods for LPA; SSMLL methods for IPA) is unreasonable, as this approach is influenced by the inherent performance of the methods themselves. Our results showed that IPA outperformed LPA. To further verify this conclusion, we used two SOTA SPML methods that were designed for LPA annotations for comparison. BoostLU and PLC are two recent methods, whose codes are available online.
>
> **Q3**: Do the paper's conclusions still apply to other contemporary algorithms, e.g., SARB[1], DualCoOp++[2], HST[3]?
>
> **A3**: Based on our experiments, the conclusion still holds for these methods. In fact, the conclusion in the paper is independent of the specific comparing methods used. It indicates that the completeness of annotations is more important than the number of annotated examples, as the other reviewer has pointed out "quality over quantity." In our experiments, when applying SOTA methods for LPA while only using a simple pseudo-labeling method for IPA, we found that the performance trained on IPA annotations still significantly outperformed that trained on LPA annotations, with a substantial margin. These results convincingly validated our conclusion. In Section 5, we disclosed that the fundamental reason behind this conclusion, which is that with complete annotation, we obtain the complete co-occurrence relationships, which are crucial for MLL.

---

> > ### Comment · Reviewer_HFP9 · 2024-11-26
> > **Official Comment by Reviewer HFP9**
> >
> > After thoroughly reviewing the responses and revisions, I believe this manuscript is suitable for acceptance by the conference.
> >
> > Lastly, is there a specific timeline for releasing the related code?

---

> > > ### Author Response · Authors · 2024-11-26
> > >
> > > Thank you for your appreciation. We will release the code after the paper is accepted, which will include the data annotation platform, mannually-annotated results and comparison pipeline.

---

### Official Review · Reviewer_aPeF · 2024-10-28

**Soundness:** 2
**Presentation:** 3
**Contribution:** 2
**Rating:** 5
**Confidence:** 5

**Summary:**

This paper compares label-level partial annotation (LPA) and instance-level partial annotation (IPA) in multi-label learning tasks to determine which is more cost-effective. The authors manually annotated MSCOCO images using both methods and found that IPA, despite annotating fewer examples, yielded significantly better model performance than LPA. The paper suggests IPA's superiority is due to its preservation of label co-occurrence relationships, which helps models capture correlative patterns.

**Strengths:**

1. The study of the cost-effectiveness of different labeling methods sounds quite interesting and has significant guidance and meaning for industry applications.

2. The experimental results are relatively comprehensive.

**Weaknesses:**

1. The paper only discusses two labeling methods. Can other labeling methods be included for discussion, such as in reference [1]?

2. LPA adopts a single positive label labeling method, which is not the traditional partial label setting [2]. If it were the traditional partial label setting which is more practical than the single positive label setting, would the analysis and conclusions of this paper still hold?

3. Defining the cost of different labeling methods is highly uncertain because, in the actual labeling process, in addition to process design, the proficiency and fatigue level of the labelers must also be considered, which can cause uneven costs. How did the authors consider this issue?

[1] Shen L, Zhao S, Zhang Y, et al. Multi-Label Learning with Block Diagonal Labels. ACM Multimedia 2024. 2024.

[2] Chen T, Pu T, Liu L, et al. Heterogeneous semantic transfer for multi-label recognition with partial labels. International Journal of Computer Vision, 2024: 1-16.

**Questions:**

See the weakness above.

---

> ### Author Response · Authors · 2024-11-20
>
> Thanks for your constructive comments. We are glad that you considered our work “sounds quite interesting, has significant guidance and meaning”. We are glad to answer all your questions.
>
> **Q1**: The paper only discusses two labeling methods. Can other labeling methods be included for discussion, such as in reference [1]?
>
> **A1**: The annotation strategy used in our paper (see Section 3.1) is very similar to the idea in [1], which divides the original classes into multiple super-classes (called blocks in [1]). In our paper, to annotate more instances using the LPA methods, we only annotate one label per image; while in [1], it annotates all the labels within the super-classes. We have included the discussion about [1] in the rebuttal version and marked them as blue. Considering different partially-annotating strategies is really an interesting future direction. We will include the annotation strategy proposed in [1] in the revised version.
>
> **Q2**: LPA adopts a single positive label labeling method, which is not the traditional partial label setting [2]. If it were the traditional partial label setting which is more practical than the single positive label setting, would the analysis and conclusions of this paper still hold?
>
> **A2**: Compared to IPA, the largest advantage of LPA is that it allows for annotating more examples. By only annotating one label per instance, we can obtain the largest number of annotated examples given a fixed annotation cost.
>
> Generally, there exists a trade-off between annotation quantity (the number of annotated examples) and annotation quality (the number of annotated labels per instance). If we annotate more labels per instance, it means that fewer examples will be annotated given the same annotation cost. This implies that the advantage of LPA would gradually diminish. There may exist a balance point where we can achieve the best model performance with a fixed level of annotation quality and annotation quantity. Finding this balance point is neither feasible nor meaningful. One thing is certain: even if this balance point is not complete annotation, it will be very close to complete annotation as done by IPA. This is because we can obtain complete co-occurrence relationships only via complete annotation, which has proven to be fundamental factor for MLL as discussed in Section 5. Including more ablation experiments on this trade-off is indeed an interesting future direction and would make our conclusions more convincing. We will include the discussion in the future version.
>
>
> **Q3**: Defining the cost of different labeling methods is highly uncertain because, in the actual labeling process, in addition to process design, the proficiency and fatigue level of the labelers must also be considered, which can cause uneven costs. How did the authors consider this issue?
>
> **A3**: To reduce the annotation uncertainty caused by the proficiency level and fatigue of the annotators, we had three annotators annotate over 2,000 examples using each partially-annotating method during the actual annotation process in order to estimate the annotation cost. By using multiple annotators and annotating a large number of data, we can minimize the randomness introduced by the mentioned two factors to the great extent.

---

### Official Review · Reviewer_Ef9E · 2024-11-01

**Soundness:** 3
**Presentation:** 3
**Contribution:** 1
**Rating:** 3
**Confidence:** 5

**Summary:**

Becasue fully annotating multi-label datasets is often impractical, this paper focuses on partial annotations. There are two primary types: (1) label-level partially-annotating (LPA), which annotates only a subset of the labels for each instance. (2) instance-level partially-annotating (IPA), which annotates a subset of the instances. This paper empirically evaluates LPA and IPA at the same annotation cost. Extensive experiments indicate that IPA preserves co-occurrence relationships, resulting in better performance.

**Strengths:**

This paper introduces three distinct variants of LPA, which have different randomness.

This paper compares LPA and IPA, finding that IPA performs better given the same annotation cost.

**Weaknesses:**

This paper explores two annotation settings that deviate from the mainstream partial multi-label setting. Please see Table 1 of DualCoOp: Fast Adaptation to Multi-Label Recognition with Limited Annotations.

A major issue in partial multi-label is the infeasibility of complete labeling due to the high number of categories, leading to potential omissions and errors. However, IPA annotates all labels for selected images, meaning it does not fully address such issue. Additionally, this work does not introduce a new approach.

Partial multi-label settings extend beyond LPA and IPA. Most mainstream methods annotate only a subset of labels per image, a topic this study does not discuss or analyze.

Table 1 should include fully labeled experimental results.

The benchmark used is limited. commonly datasets like VOC, NUSWIDE, and CUB should be included for comparison.

**Questions:**

Please see the weaknesses above.

---

> ### Author Response · Authors · 2024-11-20
>
> Thanks for your great efforts in reviewing our paper. We are glad to solve all your concerns.
>
> **Q1**: A major issue in partial multi-label is the infeasibility of complete labeling due to the high number of categories, leading to potential omissions and errors. However, IPA annotates all labels for selected images, meaning it does not fully address such issue.
>
> **A1**: In the practical annotation process, we adopt the annotation method that has been used for annotating MS-COCO (see Section 3.1). This method first divides the original 80 classes into 12 super-classes, then determines whether a super-class is present, and subsequently annotating the specific classes within the present super-class (see Figure 7 in Appendix). This method can solve the problem of incomplete annotation due to the large number of classes, as we only need to focus on a few classes within a specific super-class. Meanwhile, another advantage of this method is that it allows annotators to skip super-classes that are clearly absent, significantly reducing the annotation cost.
>
> **Q2**: This work does not introduce a new approach.
>
> **A2**: Our main contribution is to compare the cost-effectiveness of two mainstream kinds of partially annotating methods (LPA and IPA), that is, which type of partial labeling approach can achieve better model performance at the same labeling cost. To achieve this goal, we conducted extensive and comprehensive experiments and ultimately concluded that IPA is significantly more cost-effective than LPA. Finally, we disclose that complete co-occurrence relationships are the key reason why IPA is more cost-effective from the perspective of causal inference.
>
> **Q3**: Most mainstream methods annotate only a subset of labels per image, a topic this study does not discuss or analyze.
>
> **A3**: Compared to IPA, the largest advantage of LPA is that it allows for annotating more examples. By only annotating one label per instance, we can obtain the largest number of annotated examples given a fixed annotation cost.
>
> Generally, there exists a trade-off between annotation quantity (the number of annotated examples) and annotation quality (the number of annotated labels per instance). If we annotate more labels per instance, it means that fewer examples will be annotated given the same annotation cost. This implies that the advantage of LPA would gradually diminish. There may exist a balance point where we can achieve the best model performance with a fixed level of annotation quality and annotation quantity. Finding this balance point is neither feasible nor meaningful. One thing is certain: even if this balance point is not complete annotation, it will be very close to complete annotation as done by IPA. This is because we can obtain complete co-occurrence relationships only via complete annotation, which has proven to be a fundamental factor for MLL as discussed in Section 5. Including the discussion on this trade-off is indeed an important future direction and would make our conclusions more convincing. We will include more ablation experiments in the future version.
>
> **Q4**: Table 1 should include fully labeled experimental results.
>
> **A4**: Thanks for your suggestions. We have included the results of full-supervision in Tabel 1.
>
> **Q5**: The benchmark used is limited. commonly datasets like VOC, NUSWIDE, and CUB should be included for comparison.
>
> **A5**: Thanks for your suggestion. When performing experiments on a dataset, in order to obtain the annotation costs for four partially-annotating methods (three LPA methods and one IPA method), the annotators need to annotate the dataset four times manually. The enormous annotation cost prevents us from performing experiments on multiple datasets. In the current version, we have chosen the most representative MS-COCO dataset for our empirical study. We will perform experiments on additional datasets in the future version.

---

### Official Review · Reviewer_KnZT · 2024-11-03

**Soundness:** 4
**Presentation:** 4
**Contribution:** 3
**Rating:** 8
**Confidence:** 5

**Summary:**

This paper addresses the challenge of costly and time-consuming annotation in multi-label learning tasks by evaluating the cost-effectiveness of two partially-annotating methods: label-level partially-annotating (LPA) and instance-level partially-annotating (IPA). Through empirical experiments on the MS-COCO dataset, the authors demonstrate that IPA significantly outperforms LPA in terms of model performance, despite requiring fewer annotated instances. The study provides insights into the benefits of preserving co-occurrence relationships in annotations, highlighting that the quality of data can outweigh the quantity in training effective models.

**Strengths:**

I like this paper. I think this paper effectively underscores "the importance of data quality over quantity in the multi-label learning domain"
, backed by robust experimental design.

The methodology for comparing partially-annotating methods and the associated annotation costs is well thought out and executed, leading to convincing results that align with the "quality over quantity" insight.

**Weaknesses:**

More of a discussion point than a weakness.

While low-quality data generally yields subpar results, previous work [1,2] has shown that large-scale partially-annotated datasets can be created without annotation costs from image-text pairs, leading to strong generalization (zero-shot performance). Consequently, pretraining on such large-scale partially-annotated data followed by fine-tuning on fully-annotated data may be an appropriate approach towards powerful tagging models.

I encourage the authors to consider exploring in the context of such larger dataset settings in future work.


[1] Tag2Text: Guiding Vision-Language Model via Image Tagging, ICLR 2024.

[2] Recognize anything: A strong image tagging model. CVPR 2024 workshop.

**Questions:**

No specific questions

---

> ### Author Response · Authors · 2024-11-20
>
> Thanks for your appreciation of our paper. We are glad that you considered our work “robust experimental design, convincing results”. I also like the phrase “quality over quantity”.
>
> **About tagging models** It is an interesting direction to explore large-scale datasets with image-text pairs for training powerful tagging models. As mentioned, the largest advantage of this solution is that it allows for the easy conversion of image-text pairs into partial-labels, thereby generating a large amount of cheap yet usable partially-annotated data. Note that regardless of whether the image-text pairs are manually annotated or generated by LLMs, they both incur corresponding costs, such as annotation costs or LLMs training costs. In the context of our paper, considering such costs is quite difficult. I believe that this is a very promising approach and aligns well with current trends for training large-scale recognition models. We will include the discussion on this approach in the future version.

---

### Meta-Review · Area_Chair_tJ3m · 2024-12-15

**Metareview:**

This paper investigates the cost-effectiveness of partial labeling in multi-label learning by proposing two methods. The manuscript received highly diverse ratings from reviewers. In multi-label datasets, not all labels contribute equally to data analysis, suggesting that completing all labels may not always be necessary—especially when training models for downstream tasks. I agree with Reviewer Ef9E that this challenge becomes particularly pronounced with large multi-label datasets, where label correlations and distributions are often difficult to characterize due to the varying personal backgrounds of annotators. Consequently, the paper requires further improvements.

**Additional Comments On Reviewer Discussion:**

There is no consistency across all reviewers, and some reviewers still have some concerns.

---

### Decision · Program_Chairs · 2025-01-22

Reject